# Personal ethical settings for driverless cars and the utility paradox: An ethical analysis of public attitudes in UK and Japan

Kazuya Takaguchi[1,2], Andreas Kappes[3], James M. Yearsley[3], Tsutomu Sawai[4,5], Dominic J. C. Wilkinson[1,6,7]*, Julian Savulescu[1,8]

1 Oxford Uehiro Centre for Practical Ethics, Faculty of Philosophy, University of Oxford, Oxford, United Kingdom, 2 The Department of Ethics, Kyoto University, Kyoto, Japan, 3 City, University of London, London, United Kingdom, 4 Graduate School of Humanities and Social Sciences, Hiroshima University, Hiroshima, Japan, 5 Institute for the Advanced Study of Human Biology (ASHBi), Kyoto University, Kyoto, Japan, 6 John Radcliffe Hospital, Oxford, United Kingdom, 7 Murdoch Children's Research Institute, Melbourne, Australia, 8 Melbourne Law School, Melbourne, Australia

* dominic.wilkinson@philosophy.ox.ac.uk

**Data Availability Statement:** All research data (questionnaire, anonymized result, analysis) are available from the OSF database (osf.io/kv6nu).

## Abstract

Driverless cars are predicted to dramatically reduce collisions and casualties on the roads. However, there has been controversy about how they should be programmed to respond in the event of an unavoidable collision. Should they aim to save the most lives, prioritise the lives of pedestrians, or occupants of the vehicle? Some have argued that driverless cars should all be programmed to minimise total casualties. While this would appear to have wide international public support, previous work has also suggested regional variation and public reluctance to purchase driverless cars with such a mandated ethical setting. The possibility that algorithms designed to minimise collision fatalities would lead to reduced consumer uptake of driverless cars and thereby to higher overall road deaths, represents a potential "utility paradox". To investigate this paradox further, we examined the views of the general public about driverless cars in two online surveys in the UK and Japan, examining the influence of choice of a "personal ethical setting" as well as of framing on hypothetical purchase decisions. The personal ethical setting would allow respondents to choose between a programme which would save the most lives, save occupants or save pedestrians. We found striking differences between UK and Japanese respondents. While a majority of UK respondents wished to buy driverless cars that prioritise the most lives or their family members' lives, Japanese survey participants preferred to save pedestrians. We observed reduced willingness to purchase driverless cars with a mandated ethical setting (compared to offering choice) in both countries. It appears that the public values relevant to programming of driverless cars differ between UK and Japan. The highest uptake of driverless cars in both countries can be achieved by providing a personal ethical setting. Since uptake of driverless cars (rather than specific algorithm used) is potentially the biggest factor in reducing in traffic related accidents, providing some choice of ethical settings may be optimal for driverless cars according to a range of plausible ethical theories.

**Funding:** This research was funded in whole, or in part, by the Wellcome Trust [203132/Z/16/Z]. For the purpose of open access, the author has applied a CC BY public copyright licence to any Author Accepted Manuscript version arising from this submission. Julian Savulescu is Visiting Toh Chin Chye Professor in Molecular Biology and Medicine in the Centre for Biomedical Ethics at the National University of Singapore. Tsutomu Sawai is funded by JSPS KAKENHI Grant (21K12908).The funders had no role in study design, data collection and analysis, decision to publish, or preparation of the manuscript.

# 1. Introduction

## 1.1 Driverless cars

Advances in motor vehicle design have reduced the devastating harm associated with traffic collisions. For example it was estimated that forward-collision warning and autonomous braking system prevented about 14% of crash fatalities in 2016 in the US [1]. However, as many as 1.35 million deaths and 50 million injuries still occur every year worldwide from road traffic accidents [2]. This is the 8th highest cause of death, and the leading cause of death for young people aged 5–29. Further technological advances, particularly the advent of automated driving technology (driverless cars) could dramatically reduce this. In Germany, the US and UK, between 90–95% of car accidents are estimated to be caused by human error or misconduct (for example, speeding, inattention, failing to give way) [3–5]. Due to the high rates of accidents caused by human error, driverless cars are believed to have a positive impact on road safety. For example, one study estimated up to 73% reduction in pedestrian crashes in Finland [6], while a US study estimated up to 90% reduction [7]. Another survey suggests that if all human-driven cars were replaced by fully automated driverless cars this could in theory prevent 30,000 lives per year in the US [8]. Considering the great reduction in number of injuries and fatalities, this would be a massive benefit for road users [9] and the wider community by saving social expenditure [10]. These advantages have motivated governments worldwide to facilitate the adoption of driverless cars, and car manufacturers and leading tech companies to compete in their development [11].

However, the development of fully automated cars raises a number of ethical questions. One such question is how such cars should be programmed to respond in the event of a collision. Although automation will potentially eradicate human error, accidents will continue to occur (for example, due to environmental factors, technological failure, or following unexpected behaviours by other road users) albeit with reduced frequency. Faced with an imminent, unavoidable collision, human drivers have limited time or ability to respond. In contrast, fully automated vehicles can be pre-programmed to respond in one or more specific ways taking into account information from the environment and context of the accident. For example, a driverless car detecting an imminent crash could seek to protect occupants of the vehicle, it could seek to avoid harming pedestrians (or other innocent "bystanders" such as cyclists), or it could aim to minimise overall casualties (save the most lives).

## 1.2 Ethical programming of driverless cars

These different programming alternatives have been debated from the point of view of ethical theory. For example, utilitarian or consequentialist approaches typically support decisions that would minimise overall numbers of deaths or injuries (i.e., save the most lives). Other approaches (often drawing on variations of the philosophical thought experiment "the trolley problem" [12]) have questioned the idea that it would be ethical to deliberately direct a vehicle in a way that would kill pedestrians or other innocent parties. The German Ethics Commission produced a report indicating (rule 9) that driverless cars should *not* sacrifice pedestrians to save occupants [13]. On the other hand, prospective users of driverless cars may wish to protect themselves and other passengers (particularly if the passengers are friends or family members). There is a question about whether such partiality to occupants of the vehicle could be justified.

In an attempt to help guide the development of driverless cars, some empirical work has examined the views of the general public on how they think such cars should respond to collisions (their "Moral Algorithm Preference"). For example, in the Moral Machine Experiment

**Table 1. Studies investigating views of the public about ethical response to collision scenarios.**

| Author | Type of study | Style of study (methodology) | Participants | Location | Dominant preference |
|---|---|---|---|---|---|
| Awad et al., 2018 [14] | Moral Algorithm Preference | Online survey | >500,000 (40 million decisions) | 233 countries | 'Save the Most', (but some global variation) |
| | | | | | Some preference for pedestrians over occupants |
| Awad et al., 2020 [21] | Moral Algorithm Preference | Online survey | 585,531 (alternative part of MME) | 233 countries | 'Save the Most', no preference for protecting passengers |
| Bergmann et al., 2018 [22] | Moral Algorithm Preference | Virtual reality simulation | 189 | Germany | 'Save the Most' |
| Bigman and Gray, 2020 [15] | Moral Algorithm Preference | Online survey | 2352+843+993 | US, UK | 'Save the Most' (but 40% chose to treat equally) |
| Bonnefon et al., 2016 [19] | Moral Algorithm Preference | Online survey | 1928 (total) | US | 'Save the Most' |
| | Purchase Preferences | | | | 'Save the Occupants' |
| Faulhaber et al., 2019 [23] | Moral Algorithm Preference | Virtual reality simulation | 189 | Germany | 'Save the Most' |
| Frank et al., 2019 [16] | Moral Algorithm Preference | Online survey | 12,000 | US, Denmark | Save pedestrian if low numbers in car (1, or 2), Otherwise 'Save the Most' |
| | | | | | (some influence of framing/perspective) |
| Kallionen et al., 2019 [17] | Moral Algorithm Preference (virtual reality/animation) | Virtual reality simulation + online survey | 184 + 368 | Germany | 'Save the Most' |
| | | | | | (some influence of framing) |
| Li et al., 2019 [24] | Moral Algorithm Preference | Virtual reality simulation | 60 | China | 'Save the Most' |
| Liu, P and Liu, J, 2021 [20] | Moral Algorithm Preference | Online survey | 580 | China | No preference for 'Save the Most' vs 'Save the Occupants' |
| | Purchase Preferences | | | | Willingness to pay is higher with 'Save the Occupants' |
| Mayer et al., 2021 [18] | Moral Algorithm Preference | Online survey | 1380 | Germany | Save pedestrian if equal numbers, otherwise 'Save the Most' |
| Pugnetti and Schläpfer 2018 [25] | Moral Algorithm Preference | Online survey | 107 | Swiss | 'Save the Most'. Equal preference for pedestrians/occupants. |
| Wintersberger et al., 2017 [26] | Moral Algorithm Preference (driving simulator) | Driving simulator | 40 | Germany | 'Save the Most' |
| | | | | | Survival rate influences their preference. From occupants' perspective, higher personal survival rates motivate to choose altruistic decisions |

\* The table indicates preferences for how respondents think driverless cars should respond (or the choice they personally would make) in the event of a collision (Moral Algorithm Preference), and which car they would actually purchase (Purchase Preference).

\*\*We searched The National Centre for Biotechnology Information PubMed in March 2022 for papers using the keywords ("Autonomous" OR "self-driving" or "driverless") AND ("public opinion" OR "preference") AND ethics. Additional papers were identified from reference lists, related papers and the authors' libraries. Only studies reporting trade-off scenarios/preferences between saving larger/smaller numbers of people, and between saving occupants vs pedestrians are included.

(MME), online responses were obtained from almost 40 million decisions from participants in more than 200 countries [14]. Respondents judged a series of hypothetical collisions. The strongest preferences were for sparing humans over non-human animals, saving more lives, and sparing the young over the old [14]. Other studies appear to confirm widespread support for saving the most lives in a collision (though often choosing to save pedestrians if there are similar numbers of occupants/pedestrians at risk) (Table 1). A follow-up study to the MME found that about 40% of respondents chose to treat groups of potential accident casualties equally when given that option (rather than saving the greatest number) [15]. Other studies have explored the way that question framing influences responses [16–18]. When instructed to adopt the perspective of a pedestrian, respondents were more likely to support driverless car responses that would endanger occupants rather than themselves [17, 18]. However, overall

these studies suggest that the public may find it acceptable for driverless cars to be programmed with algorithms designed to minimise overall casualties, leading some to suggest that cars should be programmed with an algorithm saving the most lives as a mandatory ethical setting taking this into account [9].

One challenge is that there may be significant variations between communities in the values that they would apply to driverless car collisions. In the MME study, respondents from a "Western" cluster of countries (including North America and many European countries) had a much stronger preference for saving the most lives than respondents from "Eastern" countries (including Japan, Taiwan, China, India, and many Middle Eastern countries) [14]. The Eastern cluster had a stronger preference for saving pedestrians. Thus, the above suggestion may not be appropriate to a country where public moral preferences are different.

### 1.3 Utility paradox

A further challenge is how the public's views about the moral acceptability of driverless car programming would translate into their actual behaviour. Bonnefon et al. investigated not only public moral intuition about different driverless car algorithms, but also US consumers' willingness to buy those driverless cars ("Purchase Preferences"). Three quarters of their participants indicated that driverless cars should be programmed to minimise the number of victims (preferences were particularly strong where this would save multiple lives such as 10 people) [19]. This tendency remained (though was weaker) when participants were asked to imagine a family member in the vehicle. However, when participants were asked about purchasing driverless cars, a strikingly different response was obtained–with low support for actually buying a utilitarian-programmed vehicle. This hesitation was even greater when participants were asked to imagine that their family members might use their driverless cars. Bonnefon and colleagues concluded that although people share moral intuitions supporting utilitarian algorithms, they do not want to buy such driverless cars because of concerns for their and their family's safety [19]. Likewise, a study of 580 Chinese participants indicated a preference for purchasing driverless cars that would save occupants over pedestrians, including a willingness to pay more for such a vehicle [20] (Table 1).

This suggests a potential *utility paradox* for driverless cars: driverless car algorithms that are designed to minimise collision fatalities may lead to reduced consumer uptake, higher use of non-autonomous cars and higher overall road deaths. Purchase choices may be particularly important to consider because of the very large potential difference in risk of fatal accidents with driverless cars (compared to non-autonomous vehicles).

Because of the significant potential benefit to the community by introducing driverless cars, it would be important to assess which approach would lead to the greatest uptake. It may be preferable to either adopt a different mandatory ethical setting (i.e. other than 'save the most lives') or to allow prospective consumers to choose which they would prefer (a so-called "personal ethical setting" [9]). Empirical research is needed that would compare the expected uptake rates of different forms of mandatory ethical setting, with that of a personal ethical setting. As purchase choices may be sensitive to the price of the case we also need to assess consumers' willingness to pay for their preferred algorithm. No previous studies to our knowledge have evaluated how providing a choice of driverless car algorithm would influence expected uptake of driverless cars, nor how this might vary between cultures.

### 1.4 This study

We aimed to study how the potential consumers in two countries believe driverless cars should be programmed to respond in collisions (we call this "Moral Algorithm Preference"), as well as

their hypothetical willingness to purchase these cars (we call this "Purchase Preference"). We chose to compare the UK with Japan, since previous research suggests different ethical values impacting on driverless car preferences. We compared purchasing preferences for different programming algorithms (whether mandated or provided as an option) and examined how these choices are influenced by framing effects and purchase price. Based on these data of purchasing behaviours, we attempt to seek what algorithm or which choice of algorithm would lead to the highest driverless car uptake if made available.

## 2. Methods

### 2.1 Participants

The UK Survey was conducted online from April 7 to May 15, 2021. UK residents aged over 18 were recruited via Prolific, using convenience sampling [27]. We performed the same survey with Japanese participants from December 6, 2021, to January 5, 2022, using Crowdworks (crowdworks.jp), again using convenience sampling. This project received ethical approval from the University of Oxford (the Society and Humanities Interdivisional Research Ethics Committee, IDREC9).

For the UK survey, using G*Power [28], we calculated that a sample size of 200 participants would have sufficient power (>80%) to detect small to medium differences in price sensitivity between the driverless car models. Crowdworks has an online Japanese worker pool of 4.1 million and has been previously validated for psychology experiments [29]. We sought a larger sample (300) in Japan to take into account the possibility of higher drop-out with the broad online pool of freelance workers. Participants were reimbursed pro-rata at £7.50/hour (¥180 total for survey in Japan). An attention check was included at the beginning of the survey; those who failed were excluded from analysis. In the UK survey, 190 participants took part, of whom 186 passed the attention check and provided full data were included in the analysis. In the Japanese survey, 360 participated (346 valid responses were included in analysis). All materials and the complete data can be found at: https://osf.io/kv6nu/. Regarding demographic information, In the UK sample (N = 186), 27% were male and 64% were female, while 9% were unknown. 47% were in the range of age 20–40 and 42% were aged between 40–49 or over 50, while 11% were unknown. In the Japanese sample (N = 346). 43% were male and 49% were female, while 9% were unknown. 25% were in the range of age 18–40 and 27% were aged between 40–49 or over 50, while 48% were unknown (S1 File). Demographic information was collected from the survey providers, and not linked to individual responses.

### 2.2 Purchase preference: Personal ethical setting

The questionnaire was created in English and translated into Japanese (available at: osf.io/kv6nu) (For survey flow see S4 Text in S1 File). At the beginning, we provided background information: participants were informed that driverless cars were estimated to reduce traffic accidents by 90–94% but would need to be programmed in advance how to respond to any collisions. For the purposes of the survey, they were asked to imagine that in ten years' time, driverless cars would have the same cost and same features as a regular car but would be safer. They were asked to ignore any concerns about data privacy or liability. Their initial perceived likelihood of purchasing a driverless car (on a Likert scale from 1—Not likely to 7—Extremely likely) was assessed as a baseline measure.

Subsequently, participants were given information about programming options for driverless cars' responses to collisions. Three programming algorithms were described: 'Save the Pedestrians' [always save pedestrians in a collision between driverless car and pedestrian]; 'Save the Occupants' [always save occupants of the car]; and, 'Save the Most' [always save the

greater number of people]. We included edited images from the Moral Machine study [14] to help participants understand the different algorithms. The survey then asked them which of these programming models they would prefer if they were to purchase such a car. They were given the three programming algorithms (named, and briefly described), as well as the option of 'Random choice', wherein the driverless car would randomly choose to save either occupants or pedestrians.

## 2.3 Framing effects and Purchase Preference

Next, they were asked to imagine (and to indicate their purchasing preference), first if they were buying a car in a situation in which they had a young family member who would often be a passenger, and second in a situation in which they had a young family member who would often be a pedestrian. As before, they were given the option of the three driverless car algorithms ('Save the Pedestrians', 'Save the Occupants', 'Save the Most') or 'Random Choice'.

## 2.4 Mandatory ethical setting: Purchase and Moral Algorithm Preference

In the next section, participants were asked to indicate how likely (on a scale from 1 (Not very likely) to 7 (extremely likely)), they were to buy a driverless car if all such cars were programmed with the same algorithm (i.e., mandatory ethical setting). For example, they were told that all cars were programmed to 'Save the Most'. This question was repeated to assess likelihood of purchase for each of the three algorithms ('Save the Pedestrians',' Save the Occupants', 'Save the Most').

They were lastly asked which programming model they believed should be adopted if all driverless cars were going to be programmed in the same way (their Moral Algorithm Preference), (they were given the three algorithms as well as the option of 'Random Choice'). They were reminded that they might be either occupants or pedestrians.

## 2.5 Price sensitivity

Finally, a discrete choice experiment was conducted. Participants were presented with a series of choices between two driverless cars that feature different programming models but have different price tags and asked which car they would buy. It has been estimated that most new cars in the UK sell from about £12,000 to £23,000 [30]. We used that price range, starting with £15,000. There were 36 combinations for the price tests due to the three different models and three different prices (£15,000, £19,000 and £23,000; ¥2,250,000, ¥2,850,000, and ¥3,450,000). Fig 1 shows an example question:

## 2.6 Model description—willingness to pay for the purchase

In our model, the utility difference between Car A and Car B can be described as the difference in price and the difference in safety programming:

$$\Delta U(A, B) = (Price_B - Price_A) + (Prog_A - Prog_B) \tag{1}$$

Where $Price_A$ is the price of Car A (e.g., £15000) and $Prog_A$ is the utility associated with the programming of Car A (e.g., Utilitarian).

There are three programming options: 'Save the Occupants', 'Save the Pedestrians', 'Save the Most' (utilitarian), resulting in two independent parameters. We set the utility of the 'Save the Most' option to zero, making it the reference point for the other two programming options. And we expressed the preferences for each of these programming options ('Save the Occupants' = Occ; 'Save the Pedestrians' = Ped) in units of thousands of pounds.

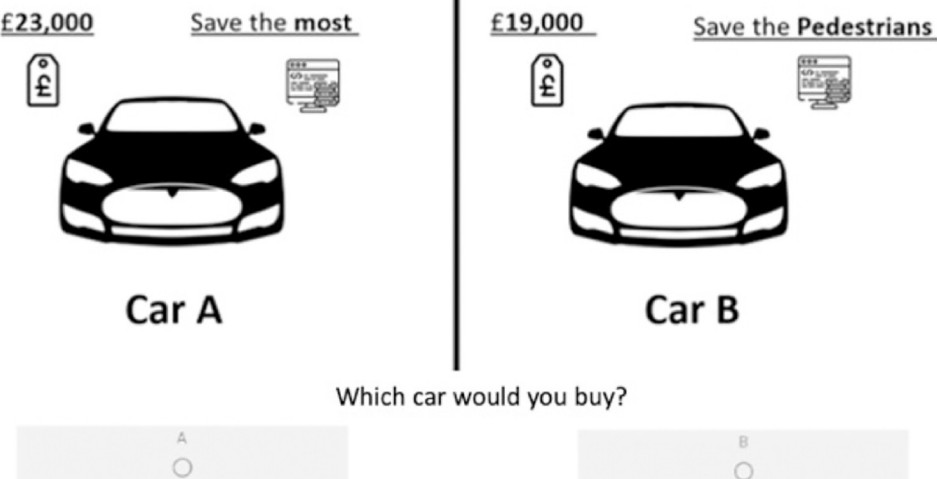

**Fig 1. An example question for the discrete choice experiment evaluating price sensitivity and preference for driverless car algorithms.**

How did we turn a utility difference $\Delta U(A, B)$ into a choice probability? Roughly, if the utility difference is large and positive, participants ought to choose A with probability ~1. And if the utility difference is large and negative, participants ought to choose B with probability ~1. To model the choice probability, we scaled the utility by multiplying by a parameter $\sqrt{\tau}$, and then applied an inverse probit transformation. This allowed us to go from the utility difference (Eq 1) to the probability of choosing A given options A and B,

$$p_{AB}(A) = probit^{-1}(\sqrt{\tau}\Delta U(A, B)) = \int_{-\infty}^{\Delta U(A,B)} \sqrt{\frac{\tau}{2\pi}} e^{\frac{\tau x^2}{2}} dx \tag{2}$$

The parameter $\tau$ represents how sensitive participants were to changes in utility, so how quickly choices switched between definitely A and definitely B as the utility varied. Lower values mean participants spent more time in the 'unsure' region.

At the individual level the choices are binary, so our choice probability that came from the model resulted in just a single yes/no response. We can model this by assuming the data comes from a single trial where the choice probability is given by $p_{AB}(A)$, in other words,

$$p(Data|\Delta U) \sim Bernouli(p_{AB}(A)) \tag{3}$$

To model this in a hierarchical way means a) every individual has their own values for the parameters *Ped*, *Occ*, $\tau$, b) These individual values can be assumed to be drawn from some population level distributions. Each of these population level distributions has an associated mean, *HPed*, *HOcc*, *H$\tau$*, and variance. More precisely,

$$Ped(i) \sim \phi(HPed, Ht), \quad Occ(i) \sim \phi(HOcc, Ht), \quad \tau(i) \sim \phi(H\tau, Ht_2) \tag{4}$$

Where $\phi(\mu, \tau)$ is the pdf of the normal distribution with mean $\mu$ and variance $1/\tau$. The idea here is that we allow for individual variation in utilities and sensitivity, but we constrain these differences so that information about the preferences of one participant is still weakly informative about the preferences of the others. *HPed*, *HOcc*, *H$\tau$*, *Ht, and Ht$_2$* are population level parameters (hyperparameters), which govern the distribution of utilities and sensitivity in the population. These are the things to be determined by the model fitting, and we assign them the

following priors,

$$HPed \sim \phi\left(0, \frac{1}{10}\right), HOcc \sim \phi\left(0, \frac{1}{10}\right), H\tau \sim \gamma\left(\frac{1}{2}, \frac{1}{2}\right), Ht \sim \gamma\left(\frac{1}{2}, \frac{1}{2}\right), Ht_2 \sim \gamma\left(\frac{1}{2}, \frac{1}{2}\right). \quad (5)$$

where $\gamma(a, b)$ is the pdf of the gamma distribution with shape parameters, a, and b.

### 2.7 Model implementation

The model was fit to the data via Bayesian methods using JAGS (Plummer, 2003 [31]), using a form of Markov Chain Monte Carlo (MCMC) [32]. Fits used three MCMC chains and 50000 MCMC samples, with a burn in of 5000 samples. Chain convergence was assessed using the $\hat{R}$ statistic, and all chains had good convergence by this metric. We report means and HDIs for the posteriors of the hyperparameters, and distributions of the Ped and Occ parameters across participants and trials.

## 3. Results

Statistical analysis was conducted using SPSS, Version 28, and computational model analysis were performed with Matlab.

### 3.1 Algorithm and Purchase Preference–which algorithm should cars have?

The largest group of participants indicated the same response when asked which algorithm they believed should be programmed into all driverless cars (Moral Algorithm Preference), and which algorithm they would prefer to purchase if able to choose (purchasing preference; personal ethical setting) in both countries (Fig 2). In the UK, the largest group selected 'Save the Most' (55.3% answered 'Save the Most' should be programmed and 45.7% answered it was their preferred algorithm), while in Japan, the largest group selected ''Save the Pedestrians' (54.6% and 56.9% respectively). A minority of participants wished to purchase the other models (UK: 'Save the Occupants' 23.4%, 'Save the Pedestrians', 23.4%; Japan: 'Save the Occupants' 15.6%, 'Save the Most' 24.6%).

While Moral Algorithm Preferences aligned with purchasing preference in each country, there was a statistically significant difference between Purchase Preferences and Moral Algorithm Preference: Moral Algorithm Preference for the UK (McNemar-Bowker (6 N = 186) = 17.60, p = .007) and for Japan (McNemar-Bowker (6) = 23.655, p = .001, N = 346).

### 3.2 The effect of framing on Purchase Preference

Next, we examined the effects of different question framing. When asked to imagine that one's family would often be in the car, a higher proportion of respondents in both countries selected to purchase 'Save the Occupants' compared to when the question was presented without such a frame (UK: 57%, McNemar-Bowker (6, N = 186) = 68.415, p < .0001; Japan: 40.2%, McNemar-Bowker (6, N = 346) = 103.814, p < .001). Similarly, there was a significant increase in preference for 'Save the Pedestrians' when asked to imagine one's family often being pedestrians compared to when the question was presented without such a frame (UK: 50%, McNemar-Bowker (6, N = 186) = 50.509, p < .0001; Japan: 74.3% McNemar-Bowker (6, N = 346) = 63.800, p < .001). Responses to both framings were different between the UK and Japan (p < .001).

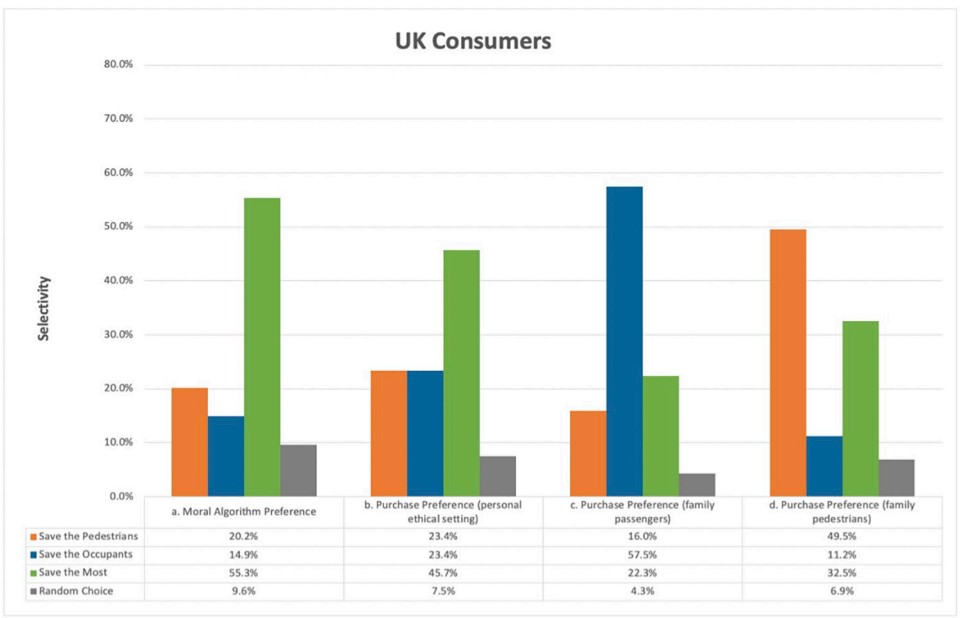

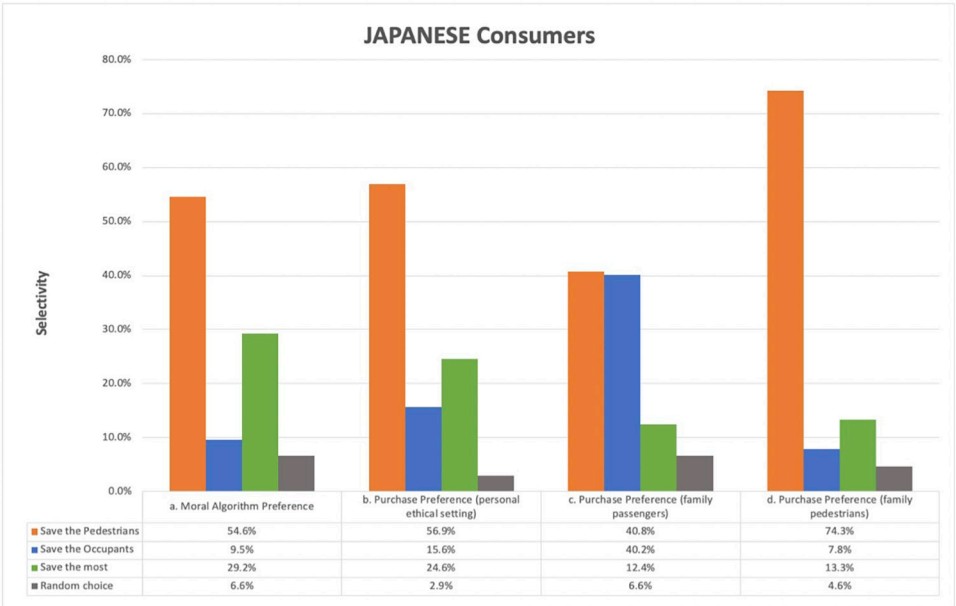

**Fig 2.** Participant preference among UK and Japanese participants for driverless car algorithms when asked: a. which algorithm should be programmed (if all cars programmed identically) b. which they would personally prefer to purchase if able to choose c. Their purchase preference if they imagined having young family who would often be passengers d. Their purchase preference if they imagined having young family who would often be pedestrians.

### 3.3 The influence of mandatory ethical settings on willingness to purchase

As a baseline question, we asked participants how likely it is that they would purchase a driverless car in ten years' time (if their current car needed replacing) without specifying any safety programming. In the UK, the median response on the 1 (not at all likely) to extremely likely (7) scale was 5, with a Mean of 5.13 (SE = .119). In Japan, the median response was 6, with a Mean of 5.66 (SE = .068). Overall, 74% of the UK participants, and 82% of the Japanese

indicated that they were likely (response >4) to buy a driverless car (neither likely nor unlikely: UK 10%, Japan 9%). Japanese participants had a higher willingness to purchase a driverless car than UK participants, t(530) = 4.166, p < .001.

Next, we examined how consumers' willingness to purchase driverless cars changes when a single algorithm is programmed for all driverless cars. For all three mandated ethical settings, there was reduced willingness to purchase compared to the baseline question indicating that some participants would only purchase if their preferred programming algorithm were available (when 'Save the Most' algorithm is only available—UK participants (M = 4.14, SE = .128, t (185) = 7.918, p < .001) and Japanese participants (M = 4.17, SE = .07, t(345) = 16.69, p < .001); when 'Save the Pedestrians' algorithm is only available—UK participants (M = 3.58, SE = .1249, t(185) = 10.925, p < .001) and Japanese participants (M = 4.77, SE = .07, t(345) = 10.72, p < .001); when 'Save the Occupants' algorithm is only available—UK (M = 4.134, SE = .123, t(185) = 7.59, p < .001) and Japanese participants (M = 4.12, SE = .093, t(345) = 17.505, p < .001)).

If a mandatory ethical setting were adopted, 47% of UK respondents would buy a car programmed to save occupants (a drop from 74% baseline) (Fig 3). 60% of Japanese respondents would purchase a car programmed to save pedestrians (a drop from 82% baseline non-specified). Overall, participants' willingness of the purchase was affected by the type of algorithm in both countries.

By tracking each participant's purchasing preference on the three algorithms, a Venn diagram was created to illustrate the distribution of consumers' preferences for those three algorithms (Fig 4). A small proportion of participants were likely to buy a driverless car whichever algorithm was available (UK 11.8%, Japan 13.6%), and a minority were not willing to purchase regardless of algorithm (UK 23.1%, Japan 21.1%). In the UK, two thirds of respondents (67.8%) were willing to purchase either 'Save the Occupants' or 'Save the Most' models. In Japan, 72.3% were willing to purchase either 'Save Pedestrians' or 'Save the Occupants'.

## 3.4 Willingness to pay: Purchase Preferences

The model was able to predict well which decisions UK and Japanese participants would make on each of the 36 scenarios (see S1 File). Examining the estimates for the population level parameters for the preference strength, we observed that for the UK population, going from a utilitarian program to one favouring pedestrians was equivalent to a price increase of £10,700, while going from a utilitarian option to one favouring occupants was equivalent to a price increase of around £3,170 (Fig 5). A different picture emerged for the Japanese population. Here, the model estimated that going from a utilitarian program to one favouring pedestrians was equivalent to a price decrease of £4,180, while going from a utilitarian option to one favouring occupants was equivalent to a price increase of around £6,470.

Comparing the population estimates (S2 Table in S1 File), the largest difference between the UK and the Japanese population was the utility assigned to 'Save the Pedestrians' programming. For the UK sample this was assigned much lower utility than the 'Save the Most' or 'Save the Occupants' options. For the Japanese sample in contrast, 'Save the Pedestrians' was significantly more attractive than either 'Save the Occupants' or 'Save the Most' options. There was also a significant difference in the utility assigned to the 'Save the Occupants' programming, with the Japanese sample rating this as significantly worse than the UK sample. Finally, we also found that Japanese participants showed a greater consistency in their utility judgments (t parameter: Japan .0045 versus UK .0034) significantly higher, and a less noisy decision-making process (Tau parameter: Japan: .388 versus UK.291), indicating that Japanese preferences were more stable than the UK ones.

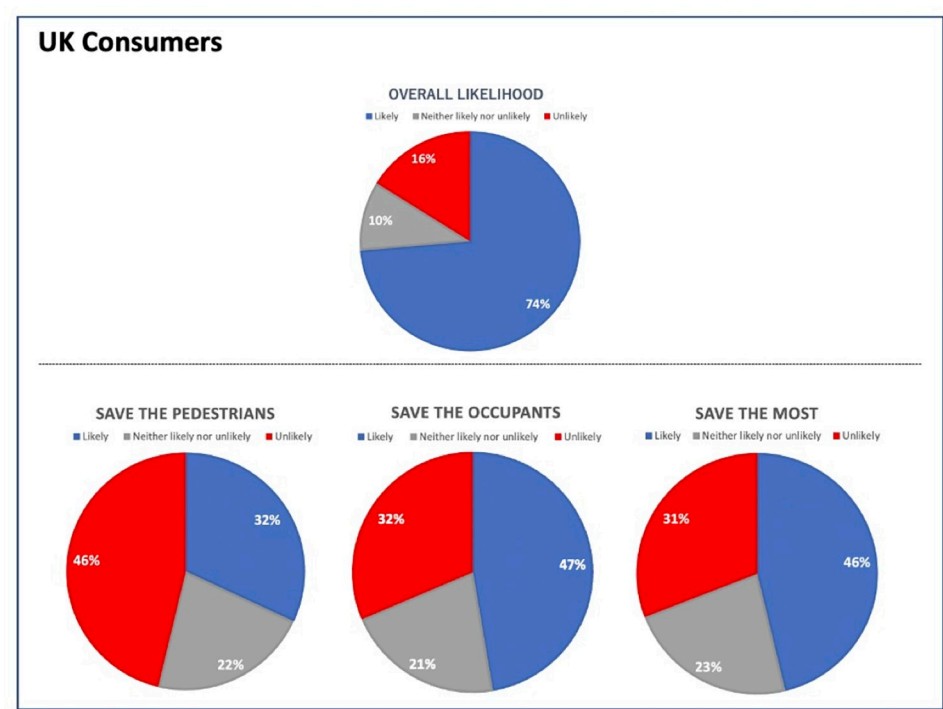

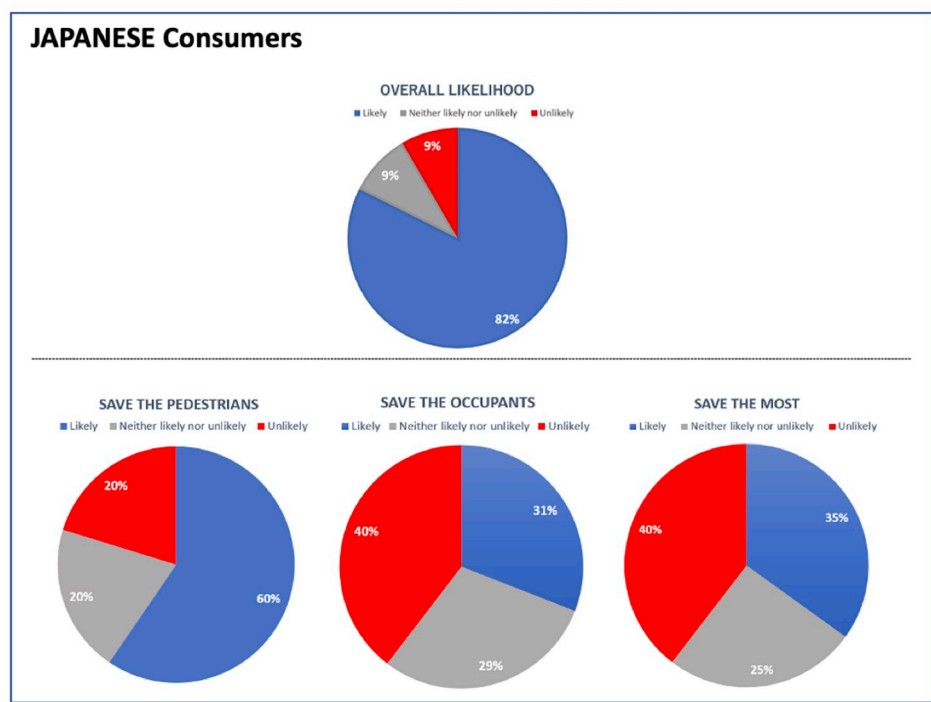

**Fig 3. The overall likelihood of purchasing a driverless car and the likelihood of purchasing a driverless car if a mandatory ethical setting were used and particular algorithms were the only option available.**

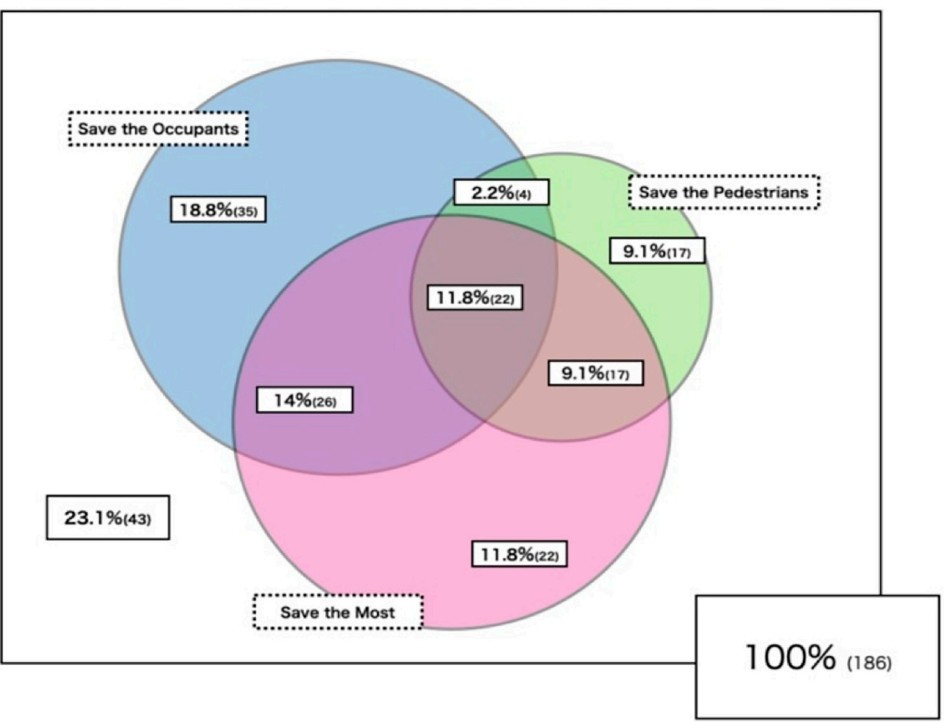

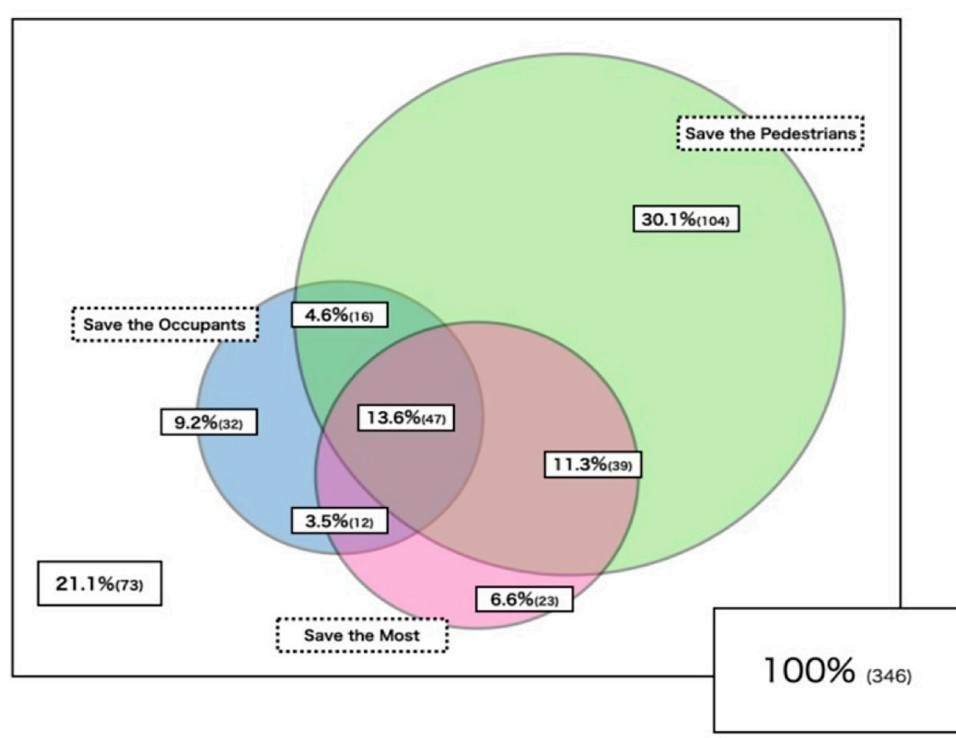

**Fig 4. Allocation of consumers' preferences for three different algorithms.** Respondents who were "unlikely" or "neither likely/unlikely" to purchase any model are indicated outside the Venn diagram (the figures in parentheses refer to the actual number of participants).

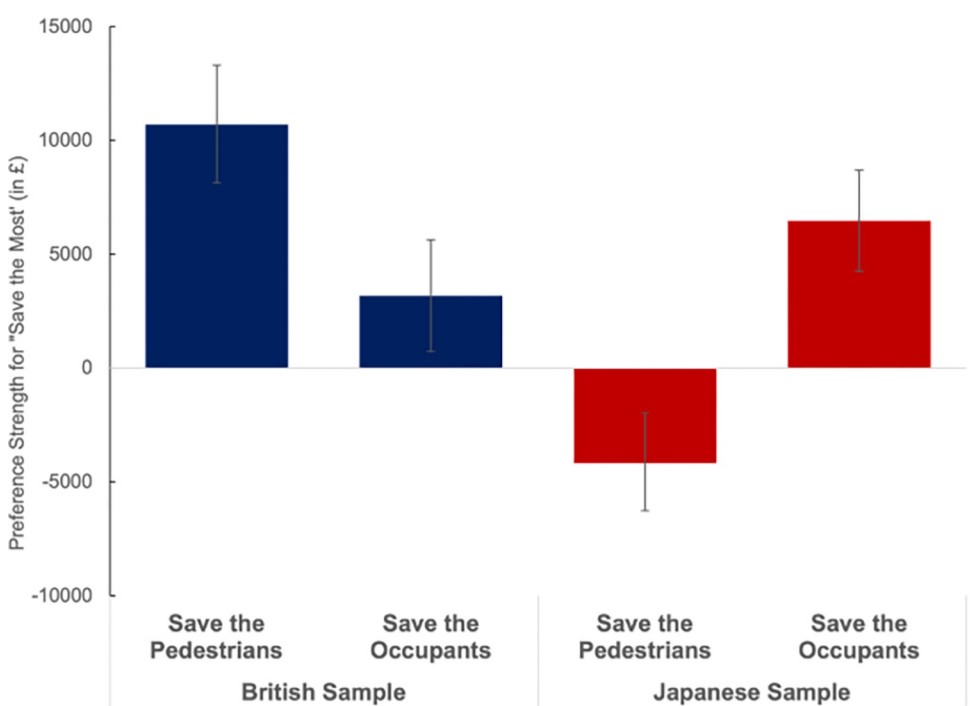

**Fig 5. Average preference strength of 'Save the Pedestrians' and 'Save the Occupants' over 'Save the Most'.** (A positive price changes means that participants preferred 'Save the Most', and would require a price discount to choose the alternative).

In each country, large individual differences existed in the UK and the Japanese Sample (S2 Fig in S1 File). When we examined the distribution of preferences values within our UK sample for 'Save the Pedestrians' over the 'Save the Most', we can see that a minority of 25 participants preferred a 'Save the Pedestrian' programming to a 'Save the Most' programming. Twenty-five participants would have paid up to £40,000 pounds more to get a car with 'Save the Most', and avoid a car with 'Save the Pedestrian' programming (S2 Fig in S1 File). In the Japanese sample, 149 participants preferred the 'Save the Most' over the 'Save the Pedestrian' safety programming. Here, 42 participants would have paid up to £10,000 to get a 'Save the Pedestrian' over a 'Save the Most' programming, and 10 participants up to £20,000 more. The most popular safety programming in each country was not endorse by everyone, and mandatory settings might deter large minorities from purchasing driverless cars. Even more pronounced is this effect when we examine the distribution of preference values in both samples for 'Save the Occupants' over 'Save the Most'. Here, we can see that in the UK sample, 65 participants preferred such programming over 'Save the Most', and in the Japanese sample, 105 participants preferred 'Save the Occupants' over 'Save the Most'.

## 4. Discussion

In this international online survey, we found striking differences between potential consumers in the UK and Japan in their preferences for the programming of driverless cars and their hypothetical purchase. A majority of UK participants (55.3%) supported the programming of driverless cars to 'Save the Most' lives in a collision. In contrast, a majority of Japanese respondents (54.6%) supported algorithms that would prioritise the saving of pedestrians. Purchasing preferences in both countries were highly sensitive to framing with shifts (in favour of saving

family members) when asked to imagine that their family members would often be either occupants or pedestrians. Importantly, in both countries, willingness to purchase was lower when ethical settings were mandated to a single algorithm, compared to baseline willingness to purchase and compared to the proportion willing to purchase when given options of driverless car algorithms. Respondents from the UK were divided (a similar proportion being willing to buy a car programmed to 'Save the Most' and to 'Save the Occupants'), while a majority in Japan were willing to purchase a driverless car that would prioritise pedestrians. In both countries, price sensitivity tracked general preferences, and participants generally placed a significant price premium on their personally preferred algorithm.

## 4.1 Cultural variation in values applied to driverless cars

The overall Moral Algorithm Preference identified in our study is consistent with previous studies. Surveys (largely in European and North American populations) have generally supported driverless algorithms that would save the most lives (Table 1), and this was described as a globally shared preference in the MME [14]. We observed a similar pattern in the responses from our UK respondents. However, within the MME, there was a significant difference between respondents from a 'Western' cluster of countries and those from an 'Eastern' cluster [14]. Our finding that Japanese respondents had a much stronger preference for prioritising pedestrians is consistent with that prior global survey. It may reflect communitarian values in Japan and strong senses of social responsibility and conformity [33]. It is also possible that the Japanese preference reflects recent media attention and social concern about accidents involving elderly drivers and pedestrians [34], or lower levels of vehicle ownership in Japan. Further studies are needed to know the reasons for Japanese purchasing behaviours.

## 4.2 Algorithm versus Purchase Preferences

We found a difference between Moral Algorithm Preferences (which algorithm they think should be programmed) and purchasing preferences. However, such purchasing preferences (in the absence of family framing) generally tracked participants' views about what should be programmed overall (Fig 2). This is somewhat in contrast to Bonnefon et al., who found a marked fall in support for a utilitarian algorithm when participants were asking about purchase [19]. However, we found that Purchase Preferences were highly sensitive to question framing. Past studies asking about general Moral Algorithm Preference have found some effect of framing perspective on responses (i.e., asked to imagine or respond as pedestrian vs occupant of car), but generally preserved utilitarian responses. We found much stronger preferences when participants were asked to imagine family members involved in accidents. It may be that our questions about Purchase Preferences are more individually directed, and therefore more sensitive to personal circumstances. Alternatively, it may be that the wording of questions in our survey primed respondents to alter their responses (possibly in a perceived socially desirable direction).

## 4.3 Mandatory ethical setting

We were interested to explore the impact of a mandatory ethical setting on overall willingness to purchase driverless cars. In Bonnefon et al.'s study, US consumers indicated aversion to governmental regulation on what algorithm to be programmed into driverless cars, which was pointed out as a concern for mandatory ethical setting. Although we did not ask the UK and Japanese consumers' attitude towards governmental regulation, our data suggest that mandatory ethical settings discourage consumer uptake since for many consumers', their decisions whether or not to buy driverless cars were contingent on which accident algorithm is available.

For example, from our UK data, a mandatory ethical setting with any type of algorithm reduced by 20% the number of consumers willing to buy driverless cars. It appeared that the largest number of respondents would buy a driverless car if given the option of choosing between the three different algorithms, or what we call a personal ethical setting. Very few respondents indicated a preference for abdicating responsibility by having the car randomly choose to save either pedestrians or occupants.

## 4.4 The utility paradox and the ethics of personal ethical settings

Overall, our results appear to support what we have labelled the 'utility paradox' in driverless car algorithms. In our sample, across two countries, algorithms designed to 'Save the Most' lives would actually lead to lower uptake of driverless cars and thus save fewer lives in practice, given the ability of driverless cars to reduce trade-off accidents in total. In Japan, that was because most respondents preferred driverless cars to prioritise pedestrians. In the UK, respondents were divided as to which algorithm they would prefer. Any mandated ethical setting attracted a smaller proportion of hypothetical consumers than those who indicated (without an algorithm specified) they were willing to purchase a driverless car.

This finding suggests that making driverless cars available with a personal ethical setting may yield a higher uptake and consequently have the greatest overall impact on road casualties, at least in the first instance. This is a libertarian option [35], but would also be predicted to have the greatest utility (and therefore be supported by utilitarianism). As some of the authors have argued elsewhere, from behind a veil of ignorance, it is rational to select the policy which saves the most lives (including those of the individual and family members) [36]. Thus, a broad contractualist approach would also support making driverless cars available with a personal ethical setting when they are introduced into general circulation. From this point of view, a personal ethical setting has the advantage that while it would be supported by utilitarians and contractualists by maximising driverless car uptake, it would also be aligned with liberalism by providing individual value-based choices.

This argument in favour of a personal ethical setting is dependent on purchasing choices, and our survey suggests that such choices may be susceptible to context and framing effects. It may be more of an advantage for some countries than others (in our Japanese participants, a 'Save the Pedestrians' mandatory setting appeared to have high support). One way for governments to increase uptake may be to subsidise the cost of driverless cars in general, or of particular algorithms (for example, in a way similar to subsidies for electric cars). For example, our willingness to pay analysis suggests that in Japan, a price subsidy of £4000 might be required to shift preferences to a 'Save the Most' algorithm. Another possibility (which would merit further study) may be to adapt algorithms to include a combination of values (for example, prioritising car occupants if passengers in the car, and similar numbers to pedestrians, while otherwise aiming to 'Save the Most'). Yet another option would be to introduce driverless cars with a personal ethical setting in the initial stage, then to shift to a (e.g. 'Save the Most') mandated ethical setting once non-automated cars have been largely or entirely replaced.

There are other ethical considerations for a personal ethical setting. One concern may be that use of such an option would lead owners of cars to be morally responsible (and potentially liable for damages) in the event of collision fatalities, perhaps analogously to driving beyond the speed limit [37]. It may be that insurance costs would be higher for those who do not choose to purchase 'Save the Most' algorithms [37]. Allowing individuals to choose their ethical setting might be socially divisive or lead to stigma. On the other hand, a personal ethical setting allows consumers to exercise their freedom and autonomy. Given a range of different plausible ethical responses (particularly where there is not a large difference in the number of

people under threat), a libertarian response [38] might allow individuals to incorporate their personal values (within limits) into their vehicle programming. It is potentially important to allow individuals to take moral responsibility for their actions when using artificial intelligence. The Japanese Cabinet Secretariat accepted this concept and pointed out that "when using AI, people must judge and decide for themselves how to use it" [39]. That would allow individuals to choose to sacrifice their own (and their passengers) to save the greatest number of lives in a collision. Individual autonomy arguably should be respected and is especially crucial in a situation where the person's life is at stake.

One objection is that autonomy can be curtailed in the public interest or for the sake of public health. Yet our research indicates a happy convergence of personal autonomy and public interest in the UK and Japan. Based on plausible assumptions about the high-safety level of driverless car system, the rarity of trade-off accidents and contingent on purchasing behaviors, the most lives would be saved if people were allowed to express their autonomy (unless governments banned non-autonomous vehicles and mandated an algorithm of 'Save the Most Lives').

### 4.5 Limitations and future directions

There are some limitations to the generalisability of these results. Online surveys have inherent selection biases. Although we used participants pools that have previously been validated in behavioural research, those who completed our surveys may not be representative of the wider population. (On the other hand, a younger, internet-using audience may overlap with the market for driverless cars). While the sample size was similar to many previous studies in this area (Table 1), a larger sample would have allowed more confident estimates of population attitudes. Only those who passed an attention check were included in the analysis, but, some of those included in the analysis may not have given careful thought to each question. This might be a problem especially in the discrete choice experiment.

As we found in our survey, respondents may be susceptible to the framing of the questions, and the specific wording of our study may have influenced results. (However, the fact that the global preferences among UK and Japanese respondents were consistent with those seen in the MME [14] and other related studies [40], supports the validity of responses). We were interested in the impact of driverless car algorithms on purchase preferences. However, stated intentions may not represent actual purchase choices. Participants may change their mind in the future or fail to imagine what they will actually do in the real situation.

### 5. Conclusions

Fully automated driverless cars are not yet commercially available. But the results of this study provide some insights that may be important for their regulation and development. Public views about the safety and ethics of driverless cars will be crucial for their acceptance. It may be that how such cars are programmed to respond may need to differ between countries–depending on the prevailing values of the community. It may also be that permitting some ethical choice in driverless car programming would be valuable.

The programming of driverless cars is fundamentally an ethical decision. We must take responsibility for the goals of such programming, whether it is saving the greatest number, or considering other factors such as age, responsibility, etc [40]. Some kind of democratic process preferably informed by ethical procedures [41] is necessary to arrive at how lives should be valued. Regulation and mandating of programming and purchase of driverless vehicles to achieve these ethical goals (once they have been derived) may be justifiable. However, in the absence of mandated replacement of non-automated vehicles, it is essential to explore the psychosocial effects of policy on purchasing behaviour because the most important factor affecting the well-

being and autonomy of road users and those affected by road use is likely to be uptake of driverless cars. If we were to minimise casualties overall, it might not be wise to adopt a 'Save the Most' algorithm as a mandated ethical setting because of the potential to decrease driverless car uptake (the utility paradox). We have argued that allowing freedom of choice of ethical setting in driverless cars would potentially be supported by utilitarian, contractualist and libertarian ethical theories. It may be the most ethical policy.

## Supporting information

**S1 File. It consists of all the supporting files for this manuscript.**
(DOCX)

## Acknowledgments

We would like to show our gratitude to Professor Tony Hope (an emeritus fellow of St Cross college at University of Oxford) for sharing his wisdom with us. Our discussion about connecting philosophical debates and empirical research was enriched by his critical insights.

## Author Contributions

**Conceptualization:** Kazuya Takaguchi.

**Data curation:** Andreas Kappes.

**Formal analysis:** Kazuya Takaguchi, Andreas Kappes, James M. Yearsley.

**Funding acquisition:** Julian Savulescu.

**Investigation:** Kazuya Takaguchi, Tsutomu Sawai.

**Methodology:** Andreas Kappes.

**Project administration:** Dominic J. C. Wilkinson.

**Resources:** Tsutomu Sawai.

**Software:** Andreas Kappes.

**Supervision:** Dominic J. C. Wilkinson, Julian Savulescu.

**Validation:** Dominic J. C. Wilkinson.

**Visualization:** Kazuya Takaguchi.

**Writing – original draft:** Dominic J. C. Wilkinson.

**Writing – review & editing:** Kazuya Takaguchi, Dominic J. C. Wilkinson, Julian Savulescu.

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
