## [Decision Letter · Decision Letter 0]

23 Aug 2022

PONE-D-22-20081Personal ethical settings for driverless cars and the utility paradox: an ethical analysis of public attitudes in UK and JapanPLOS ONE

Dear Dr. Wilkinson,

Thank you for submitting your manuscript to PLOS ONE. After careful consideration, we feel that it has merit but does not fully meet PLOS ONE’s publication criteria as it currently stands. Therefore, we invite you to submit a revised version of the manuscript that addresses the points raised during the review process.

We look forward to receiving your revised manuscript.

Kind regards,

Xiaoqiang ‘Jack’ Kong

Academic Editor

PLOS ONE

Journal Requirements:

"I have read the journal's policy and the authors of this manuscript have the following competing interests:Julian Savulescu is a Partner Investigator on an Australian Research Council grant LP190100841 which involves industry partnership from Illumina. He does not personally receive any funds from Illumina. He is an ethics consultant for Avon Cosmetics Ltd (2021-2023) and a Bioethics Committee consultant for Bayer"

Reviewers' comments:

Reviewer's Responses to Questions

**Comments to the Author**

1. Is the manuscript technically sound, and do the data support the conclusions?

Reviewer #1: Yes

Reviewer #2: Yes

2. Has the statistical analysis been performed appropriately and rigorously? 

Reviewer #1: Yes

Reviewer #2: N/A

3. Have the authors made all data underlying the findings in their manuscript fully available?

Reviewer #1: Yes

Reviewer #2: Yes

4. Is the manuscript presented in an intelligible fashion and written in standard English?

Reviewer #1: Yes

Reviewer #2: Yes

5. Review Comments to the Author

Reviewer #1: This study conducted two online surveys in UK and Japan to investigate the potential “utility paradox” of fully automated vehicles (AVs). The authors examined the purchase preference under different programming settings, including mandated and optional. Besides, the framing effect and the price sensitivity were also analyzed and discussed. The results of this paper may help the government or automobile manufacturers better understand the public’s attitude to the ethical issues of Avs. This paper is clear, interesting, and meaningful. However, the methods could be written in more detail. There are some suggestions are listed in the following:

1. In the introduction, the authors said that the 25% reduction in incapacitating injuries in the US is due to advanced motor vehicle design. However, the reduction is also related to other factors, such as advanced traffic signal control devices, better street and highway design, or education. More directed data sources may better support the authors’ ideas that AVs could reduce traffic crashes or collisions. Papers about CAV safety and simulation have proved the conclusion.

2. The personal information of the participants, such as race, gender, or age, is not discussed in the paper. If the survey has such information, it should be presented in the paper to help readers better understand the data and prerequisites for results.

3. What kinds of sampling method is applied in the paper, random or stratified sampling? Please illustrate in section 2.1.

4. In section 2.1, the authors estimated that a sample size of 200 people would have sufficient power to detect differences. The authors should illustrate the reasons why the authors make such assumptions. If some references could support your assumption, please cite them.

5. In section 2.5, the authors mention three different prices, but the reason why these three different prices are selected is not well-discussed. The determination of price setting and difference should be present in the paper.

6. In transportation engineering, the most common model for mode choice is the logit model. Please better explain why a normal distribution with a mean of 0 and variance of 1/τ is selected instead of the logit model.

7. In section 2.6, please explain the specific meaning of each population-level parameter, and how to tune the hyperparameters that should be presented in the paper.

8. In section 2.7, what is the meaning of chains and sample here? I think it should explain in more detail. Otherwise, the readers may confuse it with the survey sample (UK: 186, Japan: 346)

9. In section 4.1, the authors mentioned a significant difference between respondents from different countries. The finding of Japanese respondents is well-presented, but the finding of UK respondents is not found.

Overall, this paper is of good quality, I recommended this paper with minor revision.

Reviewer #2: Comment 1：

Page 5, line 4-5.

The introduction could be updated with newer references. What have been the injuries or fatalities trends in recent years?

Comment 2：

Table 1. The table should be structured clearer. Things are not described in the text should not be retained in the table, like “virtual reality” in column Type of study, what does that mean? I can only guess that is the methodology setting used in the study.

Comment 3:

Page 19, line 1-2.

Is there any supplementary material that can support the usage of normal distribution? From my point of view, logistic regression is more used for binary selection and the transformation from utility to probability. You may need to provide or cite some materials to support your model selection.

Comment 4:

Page 19, Equation (4).

What is HDri? Is it a typo?

Comment 5:

Page 21, line 15.

The “same pattern” makes me confused. Do the responses have the same pattern in both country? Since the comparison after this sentence is between UK and Japan. But in UK, “save the most” was the mostly selected, and “save the pedestrians” is the most selected in Japan, which should not be defined as the same. Or it means the two preferences have the same response pattern in each country? You should clarify this.

Comment 6:

Fig 2.

The quality of figures downloaded from the submission is poor and hard to see the words in the figure. The resolution of figures should meet the requirements of the publication, which is at a 300-600 ppi resolution, and not exceeding 10MB in size. Check all your figures other than just Fig 2.

Comment 7:

Page 27, section 4.1.

Is it possible that the participants in Japan survey are mostly pedestrians without a car? Did you consider the vehicle ownership in the survey?

Comment 8:

The survey structure in the methodology, the data summary and model results proposed by the authors are relatively difficult to follow. Flowcharts or tables can be helpful to follow the process.

6. PLOS authors have the option to publish the peer review history of their article (what does this mean?). If published, this will include your full peer review and any attached files.

Reviewer #1: No

Reviewer #2: No

---

## [Author Response · Author response to Decision Letter 0]

9 Sep 2022

(See also attached file)

Response to reviewers’ comments

Key: Highlighted text – changed or added text in the revised manuscript

Page/line numbers – reflect the page and line numbers in the revised manuscript

Editors

E1. Please ensure that your manuscript meets PLOS ONE's style requirements, including those for file naming. The PLOS ONE style templates can be found at 

RESPONSE: Thank you ¬– we have checked styles and file naming

E2. We note that the grant information you provided in the ‘Funding Information’ and ‘Financial Disclosure’ sections do not match. 

RESPONSE: Thank you ¬– we have revised the Financial Disclosure statement (previously it included a grant number that is not related to this project)

E3. Thank you for stating the following in the Competing Interests section: 

"I have read the journal's policy and the authors of this manuscript have the following competing interests:Julian Savulescu is a Partner Investigator on an Australian Research Council grant LP190100841 which involves industry partnership from Illumina. He does not personally receive any funds from Illumina. He is an ethics consultant for Avon Cosmetics Ltd (2021-2023) and a Bioethics Committee consultant for Bayer"

RESPONSE: We have updated a Competing Interests statement in our cover letter, as below:

““I have read the journal's policy and the authors of this manuscript have the following competing interests: Julian Savulescu is a Partner Investigator on an Australian Research Council grant which involves industry partnership from Illumina (project unrelated to this study). He does not personally receive any funds from Illumina. He is an ethics consultant for Avon Cosmetics Ltd (2021-2023) and a Bioethics Committee consultant for Bayer. This does not alter our adherence to PLOS ONE policies on sharing data and materials."

E4. Please review your reference list to ensure that it is complete and correct. If you have cited papers that have been retracted, please include the rationale for doing so in the manuscript text, or remove these references and replace them with relevant current references. Any changes to the reference list should be mentioned in the rebuttal letter that accompanies your revised manuscript. If you need to cite a retracted article, indicate the article’s retracted status in the References list and also include a citation and full reference for the retraction notice.

RESPONSE: Thank you ¬¬– we have checked the references

Reviewer 1

Reviewer #1: This study conducted two online surveys in UK and Japan to investigate the potential “utility paradox” of fully automated vehicles (AVs). The authors examined the purchase preference under different programming settings, including mandated and optional. Besides, the framing effect and the price sensitivity were also analyzed and discussed. The results of this paper may help the government or automobile manufacturers better understand the public’s attitude to the ethical issues of Avs. This paper is clear, interesting, and meaningful. However, the methods could be written in more detail. There are some suggestions are listed in the following:

 In the introduction, the authors said that the 25% reduction in incapacitating injuries in the US is due to advanced motor vehicle design. However, the reduction is also related to other factors, such as advanced traffic signal control devices, better street and highway design, or education. More directed data sources may better support the authors’ ideas that AVs could reduce traffic crashes or collisions. Papers about CAV safety and simulation have proved the conclusion.

RESPONSE: Thank you for this suggestion. 

We added direct recent data corroborating how advances in motor vehicle design prevented crash fatalities. 

“Advances in motor vehicle design have reduced the devastating harm associated with traffic collisions. For example it was estimated that forward-collision waning and autonomous breaking system prevented about 14% of crash fatalities in 2016 in the US (1) .”

“Due to the high rates of accidents caused by human error, driverless cars are believed to have a positive impact on road safety. For example, one study estimated up to 73% reduction in pedestrian crashes in Finland (6), while a US study estimated up to 90% reduction (7). Another survey suggests that if all human-driven cars were replaced by fully automated driverless cars this could in theory prevent 30,000 lives per year in the US (8).”

 The personal information of the participants, such as race, gender, or age, is not discussed in the paper. If the survey has such information, it should be presented in the paper to help readers better understand the data and prerequisites for results.

RESPONSE: Thank for this suggestion. We have added demographic data for our sample in the supporting information.

 What kinds of sampling method is applied in the paper, random or stratified sampling? Please illustrate in section 2.1.

RESPONSE: Recruitment from crowed sourced services such as Mturk and Prolific is best described as convenience sampling. We added this to Section 2.1.

“The UK Survey was conducted online from April 7 to May 15, 2021. UK residents aged over 18 were recruited via Prolific, using convenience sampling. We performed the same survey with Japanese participants from December 6, 2021, to January 5, 2022, using Crowdworks (crowdworks.jp), again using convenience sampling.”

1.4. In section 2.1, the authors estimated that a sample size of 200 people would have sufficient power to detect differences. The authors should illustrate the reasons why the authors make such assumptions. If some references could support your assumption, please cite them.

RESPONSE: Thanks for bringing this to our attention. We now write: 

For the UK survey, using G*Power(27), we calculated that a sample size of 200 participants would have sufficient power (>80%) to detect small to medium differences in price sensitivity between the driverless car models.

1.5. In section 2.5, the authors mention three different prices, but the reason why these three different prices are selected is not well-discussed. The determination of price setting and difference should be present in the paper.

RESPONSE: We chose the numbers based on the average car prices in the UK (ref). It has been estimated that most new cars sell from about £12,000 to £23,000. We used that prize range, starting with £15,000. We added this to section 2.5.

“It has been estimated that most new cars in the UK sell from about £12,000 to £23,000 [29]. We used that price range, starting with £15,000. “

1.6. In transportation engineering, the most common model for mode choice is the logit model. Please better explain why a normal distribution with a mean of 0 and variance of 1/τ is selected instead of the logit model.

RESPONSE: We thank the reviewer for bringing this to our attention. Essentially what we are describing is a probit function linking the (suitably scaled) difference in utility to the response probability. A probit function was favoured over a logit simply because it is easier to implement in a Bayesian modelling framework. We do not expect that the choice of probit vs logit would have an influence on our findings. We have added a note in the text to make this link clear.

“To model the choice probability, we scaled the utility by multiplying by a parameter √τ, and then applied an inverse probit transformation. This allowed us to go from the utility difference (Equation 1) to the probability of choosing A given options A and B

1.7. In section 2.6, please explain the specific meaning of each population-level parameter, and how to tune the hyperparameters that should be presented in the paper.

RESPONSE: We thank the reviewer for noting this. Each individual has their own value of the utility they assign to, eg, the Pedestrian favouring algorithm. These individual utilities are assumed to be drawn from some distribution at the population level with mean H_Ped and variance (1/H_t). 

So, H_Ped and H_Occ are the means of the distributions of the utilities of the Pedestrian and Occupant favouring algorithms across the population. H_t controls the heterogeneity of the utilities across the population, so larger values imply a greater degree of agreement between members of the population about the value of these utilities.

Every participant also has a parameter tau that controls their sensitivity to differences in utility between different alternatives. Larger values of tau mean a greater sensitivity. These are assumed to be drawn from some distribution at the population level with mean H_tau and variance (1/H_t2). As above, H_t2 controls the heterogeneity of these sensitivities across the population.

In terms of ‘tuning’ these parameters, the model was fit to the data using Markov Chain Monte Carlo methods, as briefly discussed in Section 2.7. The Supporting Information reports means and 95% HDIs of the posterior estimates for each of these parameters.

“The model was fit to the data via Bayesian methods using JAGS (Plummer et al., 2003), using a form of Markov Chain Monte Carlo (MCMC) [30]. Fits used three MCMC chains and 50000 MCMC samples, with a burn in of 5000 samples.”

1.8. In section 2.7, what is the meaning of chains and sample here? I think it should explain in more detail. Otherwise, the readers may confuse it with the survey sample (UK: 186, Japan: 346)

RESPONSE: These terms describe aspects of the Markov Chain Monte Carlo process that was used to fit the model. ‘Chain’ refers to the results of single run of the algorithm, multiple runs are usually performed to assess convergence. ‘Samples’ refers to the number of steps in the algorithm that have been performed. MCMC algorithms are guaranteed to converge only asymptotically, so the number of samples is typically very large. For more information about MCMC sampling we have provided a reference explaining this technique (see also response 1.7).

1.9. In section 4.1, the authors mentioned a significant difference between respondents from different countries. The finding of Japanese respondents is well-presented, but the finding of UK respondents is not found.

RESPONSE: Thank for highlighting this omission. We have added the following to this paragraph.

“Surveys (largely in European and North American populations) have generally supported driverless algorithms that would save the most lives (Table 1), and this was described as a globally shared preference in the MME(12). We observed a similar pattern in the responses from our UK respondents.”

Overall, this paper is of good quality, I recommended this paper with minor revision.

Reviewer 2

2.1:

Page 5, line 4-5

The introduction could be updated with newer references. What have been the injuries or fatalities trends in recent years?

RESPONSE: Thank you for this suggestion. We have updated the references in this section. See response 1.1

2.2：

Table 1. The table should be structured clearer. Things are not described in the text should not be retained in the table, like “virtual reality” in column Type of study, what does that mean? I can only guess that is the methodology setting used in the study.

RESPONSE: Thank you for this suggestion. We have added a column to the table indicating the methodology of the studies cited. We have also made some revisions for clarification.

2.3:

Page 19, line 1-2.

Is there any supplementary material that can support the usage of normal distribution? From my point of view, logistic regression is more used for binary selection and the transformation from utility to probability. You may need to provide or cite some materials to support your model selection.

RESPONSE: Thank you for this suggestion. We have added details to explain the methodology used. See response to Reviewer 1, point 1.6.

2.4:

Page 19, Equation (4).

What is HDri? Is it a typo?

RESPONSE: Thank you for pointing this out. We have corrected this.

2.5:

Page 21, line 15.

The “same pattern” makes me confused. Do the responses have the same pattern in both country? Since the comparison after this sentence is between UK and Japan. But in UK, “save the most” was the mostly selected, and “save the pedestrians” is the most selected in Japan, which should not be defined as the same. Or it means the two preferences have the same response pattern in each country? You should clarify this.

RESPONSE: Apologies for this lack of clarity. We meant to indicate that UK and Japan had the same pattern in the sense that the algorithm they think should be programmed corresponded with their preferred algorithm to purchase. We have edited this part to hopefully clarify the point.

“While Moral Algorithm Preferences aligned with purchasing preference in each country, there was a statistically significant difference between purchase preferences and Moral Algorithm Preference:”

2.6:

Fig 2.

The quality of figures downloaded from the submission is poor and hard to see the words in the figure. The resolution of figures should meet the requirements of the publication, which is at a 300-600 ppi resolution, and not exceeding 10MB in size. Check all your figures other than just Fig 2.

RESPONSE: Thank you – we have uploaded higher resolution versions of the figures 

Comment 7:

Page 27, section 4.1.

Is it possible that the participants in Japan survey are mostly pedestrians without a car? Did you consider the vehicle ownership in the survey?

RESPONSE: Thank you for this excellent question. We did not ask if the participants own a car or not. As you point out, Japan's tendency for wishing to spare pedestrians might relate to lower vehicle ownership (approximately 69% compared to 77% in UK). We have added this possibility in the discussion

“It is also possible that the Japanese preference reflects recent media attention and social concern about accidents involving elderly drivers and pedestrians (27), or lower levels of vehicle ownership in Japan. Further studies are needed to know the reasons for Japanese purchasing behaviours.” 

Comment 8:

The survey structure in the methodology, the data summary and model results proposed by the authors are relatively difficult to follow. Flowcharts or tables can be helpful to follow the process.

RESPONSE: Thank you for this suggestion. We have added a flow chart of the survey in the supporting information

---

## [Decision Letter · Decision Letter 1]

26 Sep 2022

Personal ethical settings for driverless cars and the utility paradox: an ethical analysis of public attitudes in UK and Japan

PONE-D-22-20081R1

Dear Dr. Dominic Wilkinson,

We’re pleased to inform you that your manuscript has been judged scientifically suitable for publication and will be formally accepted for publication once it meets all outstanding technical requirements. Both reviewers has provied few additional comments. Please reflect these comments on your latest version of manuscript! 

Kind regards,

Xiaoqiang ‘Jack’ Kong

Academic Editor

PLOS ONE

Additional Editor Comments (optional):

There are a few comments provided by reviewers! Please address them before you submit the final version.

Reviewers' comments:

Reviewer's Responses to Questions

**Comments to the Author**

1. If the authors have adequately addressed your comments raised in a previous round of review and you feel that this manuscript is now acceptable for publication, you may indicate that here to bypass the “Comments to the Author” section, enter your conflict of interest statement in the “Confidential to Editor” section, and submit your "Accept" recommendation.

Reviewer #1: All comments have been addressed

Reviewer #2: All comments have been addressed

2. Is the manuscript technically sound, and do the data support the conclusions?

Reviewer #1: Yes

Reviewer #2: Yes

3. Has the statistical analysis been performed appropriately and rigorously? 

Reviewer #1: Yes

Reviewer #2: Yes

4. Have the authors made all data underlying the findings in their manuscript fully available?

Reviewer #1: Yes

Reviewer #2: Yes

5. Is the manuscript presented in an intelligible fashion and written in standard English?

Reviewer #1: Yes

Reviewer #2: No

6. Review Comments to the Author

Reviewer #1: The authors have properly addressed the concern, but there is some feedback in the following.

1. The authors provided the demographic information of the data. It seems there are slight differences between the UK and Japan in terms of gender and age distribution. Due to the convenience sampling, such a difference is tolerable.

2. The choice of logit and probit would nearly not influence the findings, but the former assumes the random variable obeys a logistic distribution, while the latter assumes that the random variable obeys normal distribution.

Overall, I have no further comments for this paper.

Reviewer #2: Comment #1

In the revised manuscript, there are some typos on page 5 between line 4 and 5.

The "forward-collision waning" should be "forward-collision warning" and the "autonomous breaking system" should be "autonomous braking system".

Comment #2

Page 19, line 1 and line 18.

The meaning of "Occ" and "Ped" in both two lines is not clear. Italics are mostly used to indicate variables, and the use of different font styles conveys different meanings, which the author needs to clarify.

7. PLOS authors have the option to publish the peer review history of their article (what does this mean?). If published, this will include your full peer review and any attached files.

Reviewer #1: No

Reviewer #2: No

---

## [Editor Report · Acceptance letter]

20 Oct 2022

PONE-D-22-20081R1 

Personal ethical settings for driverless cars and the utility paradox: an ethical analysis of public attitudes in UK and Japan 

Dear Dr. Wilkinson:

I'm pleased to inform you that your manuscript has been deemed suitable for publication in PLOS ONE. Congratulations! Your manuscript is now with our production department. 

Kind regards, 

on behalf of

Dr. Xiaoqiang ‘Jack’ Kong 

Academic Editor

PLOS ONE